# Positive rhizosphere priming accelerates carbon release from permafrost soils

Nina L. Friggens [1] ✉, Gustaf Hugelius [2], Steven V. Kokelj[3], Julian B. Murton [4], Gareth K. Phoenix [5] & Iain P. Hartley [1]

Thawing permafrost soils are predicted to release substantial amounts of carbon by 2100. In addition to this, warming-induced active-layer deepening and increased rooting depth may result in further carbon losses from previously-frozen soil by stimulating microbial communities through fresh carbon inputs inducing positive rhizosphere priming. While models based on temperate data predict significant permafrost carbon loss through rhizosphere priming, data from permafrost soils are lacking. Here, we provide direct evidence of live plant-induced positive rhizosphere priming in permafrost and active-layer soils across diverse soil types from Arctic and Subarctic Canada. By $^{13}CO_2$ labelling plants in a controlled environment, we show that root activity increases carbon loss from previously frozen soils by 31%. This rhizosphere priming effect persists longer in permafrost than in active-layer soils, suggesting greater vulnerability of permafrost carbon. These findings underscore the urgency of incorporating plant–soil–microbe interactions into models predicting greenhouse gas emissions from thawing permafrost.

Northern permafrost regions are one of the largest and most vulnerable stores of soil organic carbon (SOC) globally[1,2]. These frozen carbon (C) stores are threatened by rising global temperatures leading to permafrost thaw, which may release 10s of billions of tonnes of C by 2100[3,4]. Due to widespread increases in the depth of seasonal thaw as a result of warming, near-surface permafrost is thawing in many polar regions[5]. When such active-layer deepening occurs, a previously unoccupied soil volume becomes available for plant roots that invade newly-thawed permafrost[6], with experimental evidence showing that plants access available nutrients at the thaw front[7,8]. Furthermore, it has been shown that roots found in current permafrost soils grew downwards through newly-thawed soil during the early Holocene warm interval in the western Arctic, prior to refreezing within aggrading permafrost in the mid-late Holocene[9]. Importantly, modern permafrost thaw coupled with greater plant productivity[10] and increased rooting depths may increase rates of C inputs to soil in permafrost regions. Based on observations from temperate regions,

there are predictions that greater C inputs could stimulate decomposition of previously-frozen soil organic matter (SOM) by a mechanism known as the rhizosphere priming effect (RPE), but direct evidence for this effect remains limited with few empirical data on live plant-induced RPEs in permafrost environments.

The RPE represents a change in SOM decomposition induced by plant roots compared to soil without roots, and is caused by plant root activity supplying soil microorganisms with energy-rich compounds such as root exudates and litter[11]. The RPE can be positive (accelerating SOM decomposition) or negative (when root activity suppresses SOM decomposition).

Positive RPEs may occur when microorganisms utilize fresh C as an energy source to produce extracellular enzymes which facilitate the decomposition of complex organic molecules[12] to access and metabolise C and nitrogen (N). One example of this is 'N-mining', where extracellular enzymes are deployed to release nutrients from complex organic matter, and is one of the proposed mechanisms for arctic

[1]Department of Geography, Faculty of Environment, Science and Economy, University of Exeter, Exeter, UK. [2]Department of Physical Geography, Stockholm University, Stockholm, Sweden. [3]Northwest Territories Geological Survey, Government of the Northwest Territories, Yellowknife, NT, Canada. [4]Department of Geography, University of Sussex, Brighton, UK. [5]Plants, Photosynthesis and Soil, School of Biosciences, University of Sheffield, Sheffield, UK. ✉ e-mail: n.lindstrom-friggens@exeter.ac.uk

RPEs[13,14] corroborated by the absence of positive RPEs in plots dominated by N-fixing Alder[15]. This implies that RPEs may be controlled by microbial nutrient demand[16], however evidence from incubation experiments suggests that microbial N-mining is unlikely to be the sole mechanism driving positive priming[17,18]. Alternatively, RPEs may be driven by SOM quality and microbial C and energy availability[12]. Negative RPEs may occur if microbes shift to preferentially utilising fresh C inputs rather than SOM or because increased competition for nutrients between plant roots and microbes decreases microbial growth and metabolism, thereby depressing SOM decomposition[11]. These proposed mechanisms for RPEs[11] remain largely untested in permafrost soils.

Radiocarbon evidence from field studies suggests that arctic soils are vulnerable to RPEs[15,19]. Furthermore, recent modelling work indicates that RPEs may amplify overall soil respiration in permafrost-affected ecosystems by ~12% and could increase absolute soil C loss from northern permafrost regions by ~40 Pg soil C by 2100[20]. However, the model could not directly parameterise priming effects induced by live plants in Arctic or Subarctic soils due to lack of data from these regions. Research into the priming of permafrost soils has been carried out using incubation studies in soil-only systems with C supplied not through plants but as labile substrate additions[21–26]. However, this method may miss the chemical, temporal and spatial complexity of live plant-delivered root exudates[27], and lacks a key sink for nutrients, namely plant biomass. Several proposed mechanisms driving both positive and negative RPEs depend on competition for nutrients either between microbes or between plants and microbes[16,28] which likely differs fundamentally in plant–soil systems compared with soil-only systems. Therefore, there is a critical knowledge gap in understanding the genuine role that plant-derived C plays in priming decomposition of SOM and increasing C release from permafrost soils as thaw depths (and hence rooting depths) increase.

We hypothesise that i) fresh plant-C inputs into previously frozen soils through root exudation will stimulate the microbial community, causing greater mineralisation of old C from soils with roots compared to root-free controls; and ii) positive RPEs will be driven by microbial nutrient demand rather than being due directly to the energy provided by fresh C inputs.

We quantified RPEs induced by live plants growing in a range of permafrost soils, including both active layer and permafrost horizons from contrasting soils representative of large areas of northern permafrost regions. To achieve this, we developed a bespoke $^{13}$C-labelling plant-growth chamber[29] capable of i) maintaining constant $^{13}CO_2$ atmospheric enrichment of 500‰ across fixed day–night cycles for 370 days, equivalent to about five Arctic growing seasons; ii) controlling soil temperatures substantially below air temperatures, as experienced in Arctic summers; and iii) quantifying soil $CO_2$ efflux and its $^{13}$C signature from 216 experimental units. The use of $^{13}CO_2$ and soils with and without roots allows partitioning of C effluxes from soil into i) plant-derived C and ii) RPE-induced release of soil C. Our experimental design also investigated the relative importance of live roots versus root litter in controlling priming effects, as well as the role of nutrient availability in controlling the direction and/or magnitude of priming effects.

Here we show that live plant roots significantly enhance SOM decomposition in both permafrost and active layer soils, leading to net C loss through rhizosphere priming (positive RPE). Across a range of Arctic and Subarctic soils, the presence of roots increased SOM-derived $CO_2$ flux by an average factor of 1.31 over 370 days. In active layer soils, priming effects were strongest during the first 185 days before declining. In contrast, permafrost soils exhibited sustained and increasing priming effects through time. The persistence of priming in permafrost suggests that positive RPEs in thawing permafrost soils may increase C losses, amplifying the effect of permafrost thaw on climate change.

## Results

### Positive RPEs in active layer and permafrost soils

Soil types from dominant suborders of permafrost-affected soils (Gelisols, i.e., Orthels, Turbels) were collected along a transect across continuous and extensive discontinuous permafrost in northwest Canada (site descriptions in Supplementary Information). The soils developed on a variety of major Quaternary superficial deposits, including silt- and clay-rich till, sandy alluvium, silt-rich yedoma and organic-rich thermokarst-lake-basin deposits. They cover the main types of permafrost terrain (glaciated vs non-glaciated) and growth (epigenetic vs syngenetic) in the Arctic and Subarctic.

The collected permafrost and active layer soils were used to investigate rhizosphere priming effects in permafrost region soils within a temperature controlled, $^{13}$C-labelling plant-growth chamber[29]. By measuring $CO_2$ fluxes and $^{13}$C signatures from the headspace of both rooted and root-free compartments of custom-designed mesocosms (see Methods for details), we were able to partition soil C effluxes into plant-derived $CO_2$ and SOM-derived $CO_2$ thereby quantifying the RPE-induced release of soil C.

Across the wide range of permafrost and active layer soils tested here (Supplementary Fig. 1) the presence of roots caused greater fluxes of SOM-derived $CO_2$ and RPE ratios >1, resulting in net C release from both active layer and permafrost soils (Fig. 1). Our data from all soils and over the full duration of the experiment since plant germination (370 days) show that SOM-derived $CO_2$ flux from soils with roots was on average higher than from root-free soil by a factor of 1.31 (1.22–1.40 95% CI; RPE ratio; Fig. 1a). No discernible patterns were observed in the RPE data between soil types (Fig. 2).

Severing ingrown roots—thereby halting the supply of fresh root exudates—results in a drop of SOM-derived $CO_2$ flux within one month (Supplementary Fig. 3b, c), suggesting rhizosphere priming was largely driven by fresh root exudates, rather than root litter. The drop in SOM-derived $CO_2$ flux induced by root severing likely occurred sooner than one month, as root exudates are turned over and respired within days[30] and therefore likely to have a short-lived impact on SOM-derived $CO_2$ flux. However, with our monthly flux measurements we are unable to quantify precisely how quickly positive priming effects were lost. The observed drop in SOM-derived $CO_2$ flux following the severing of live roots corroborates evidence from work which found that synthetic root exudates can induce soil C loss through soil priming and mobilisation of soil C[24,27,31,32]. This experimental observation is significant for modelling the magnitude and spatial distribution of RPEs in permafrost regions, with many areas seeing significant increases in aboveground biomass (arctic "greening") and therefore fresh soil C inputs, along with areas which are browning (decline in biomass) and burning in spatially heterogeneous ways[33] with likely impacts on belowground delivery of fresh C in both space and time.

In active layer soils, positive RPEs occurred in the first 185 days (Fig. 1c, d, $p < 0.05$ on days 56 & 173), in contrast to results from incubation studies with labile substrate additions[22,23] which find no priming of active layer soils. These differing experimental results may indicate that active layer soils are vulnerable to priming by live root activity but not by the addition of labile substrates. The RPE ratio in active layer soils was 1.39 (1.21–1.57 95% CI; Fig. 1c) in days 0–185 and 1.03 (0.89–1.17 95% CI; Fig. 1c) between days 186 and 370. This change in the magnitude of RPEs over time suggests that there may only be a small pool of C in active layer soils that is vulnerable to priming. Alternatively, the result could be linked to reduced plant growth later in the experiment; less frequent clipping of aboveground biomass was necessary in the later parts of the experiment. However, we consider the plant growth explanation to be less likely because the reduction in positive priming beyond day 186 was not observed in permafrost soils, which had the same above-ground plant growth trends. In addition, headspace $CO_2$ concentrations did not decline substantially over time, indicating that root activity

remained high (Supplementary Fig 3a). In contrast, in permafrost soils positive rhizosphere priming occurs throughout the experiment (Fig. 1d, e, $p < 0.05$ for most but not all measured days), with an RPE ratio of 1.39 (1.29–1.50 95% CI; Fig. 1e), and RPE ratios greater than 1.5 observed from day 300. After day 250, permafrost soils had a significantly higher RPE ratio than active layer soils ($p = 0.0035$, F = 5.21, df = 3). The maintenance and potential increase in priming in permafrost soils later in the experiment, in contrast to the decline in priming over time in active layer soils, suggest that permafrost soils may be more vulnerable than active layer soils to positive RPEs through time. Given that the majority of soil C in northern permafrost regions is stored below depths of 30 cm, the potential for sustained positive RPEs in permafrost soils and increased mineralisation of this C could amplify the positive feedback between permafrost $CO_2$ emissions and global warming[4,34].

### Factors controlling the magnitude of positive RPEs

The greater and more persistent positive priming effects observed in permafrost soils may arise if the permafrost microbial community in older, deeper soil layers, with more degraded SOM, are more limited by labile C and energy availability[12] than active layer soils. Consistent with this explanation, rates of $CO_2$ release per unit C were lower in permafrost than active layer soils ($p = 0.029$, F = 5.395, df = 1, Fig. 2d, e). Stronger microbial C limitation has been linked to greater priming effects following [13]C-labelled substrate additions in permafrost soils[26]. The RPE ratio was significantly negatively related to cumulative SOM-derived $CO_2$ flux for all soils ($p = 0.0015$, F = 12.37, df = 29). This suggests that the greatest priming effects occurred in soils with low bioavailable SOM, consistent with stronger microbial C limitation being related to increased priming intensity.

The difference in RPEs over time between active layer and permafrost soils may alternatively or additionally be driven by the mobilisation of physico–chemically protected organic matter in mineral permafrost soils. Root exudates, specifically those containing organic acids, have been found to liberate organic matter from mineral association thereby making it microbially available[32]. As a greater proportion of SOC has been shown to be minerally-associated in deeper mineral permafrost soils than in organic-rich active layer soils, this effect could be more pronounced in permafrost soils and may explain the patterns in RPEs observed here. However, no relationships were found between RPEs and soil primary texture in this experiment (Supplementary Fig. 1d, e). Further experiments are needed to investigate these mechanisms fully and to elucidate the extent to which positive RPEs in permafrost are driven by microbial C limitation or direct action of plant derived organic acids.

The soils used here covered a wide range of %C, %N and C:N ratios (Supplementary Fig. 1a), allowing further investigation of factors that have been implicated in driving the magnitude of priming effects[12,24,35]. Positive rhizosphere priming was observed across soils with differing C contents, N contents and C:N ratios. However, the RPE ratio exhibited a significant negative power-law relationship with soil C and N content across all soils (C; $p = 0.0016$, F = 12.18, df = 29, N; $p = 0.0047$, F = 9.37, df = 29) and in permafrost (C; $p = 0.027$, F = 2.76, df = 16, N; $p = 0.021$, F = 2.90, df = 16) but not active layer (C; $p = 0.45$, F = 1.10, df = 11, N; $p = 0.35$, F = 1.27, df = 11) soils (Fig. 2a, b). Across the full experimental duration, at a given C or N content, RPEs tended to be greater in permafrost horizons, but this effect was only statistically significant for N (C; F = 3.64, $p = 0.067$, N; F = 4.42, $p = 0.044$). This indicates that organic matter in organic-rich active layer and permafrost soils may be less vulnerable to priming, which could be driven by lower microbial C

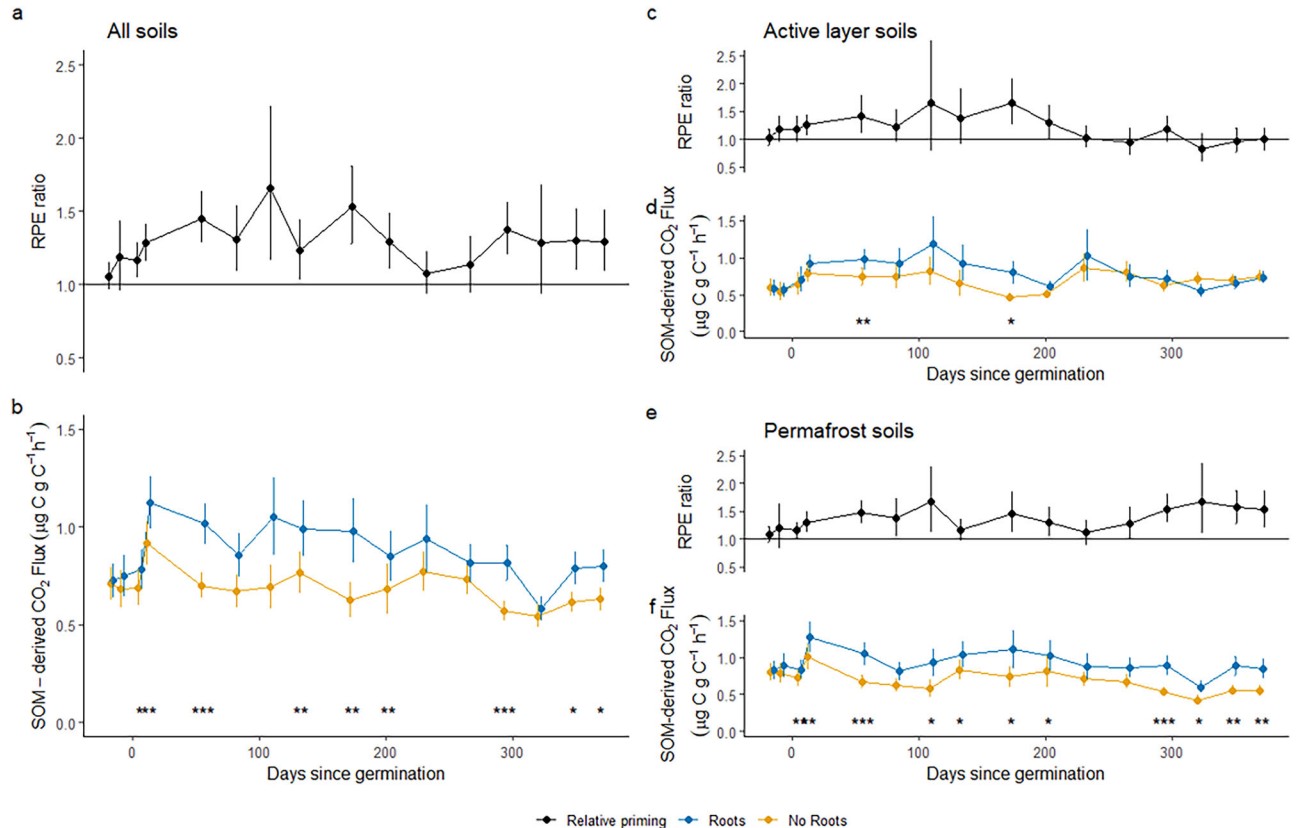

**Fig. 1 | RPE ratios and SOM-derived $CO_2$ fluxes over time.** RPE ratio with error bars showing 95% confidence intervals (top y-axis) and absolute fluxes of SOM-derived $CO_2$ with error bars showing standard error (bottom y-axis) in paired soils with and without roots in all (**a**, **b**, $n = 31$), active layer (**c**, **d**, $n = 13$) and permafrost (**e**, **f**, $n = 18$) soils over time. Significant differences between rooted and root-free soils, based on paired t-tests, are indicated with asterisks (*$p < 0.05$, **$p < 0.01$, ***$p < 0.001$).

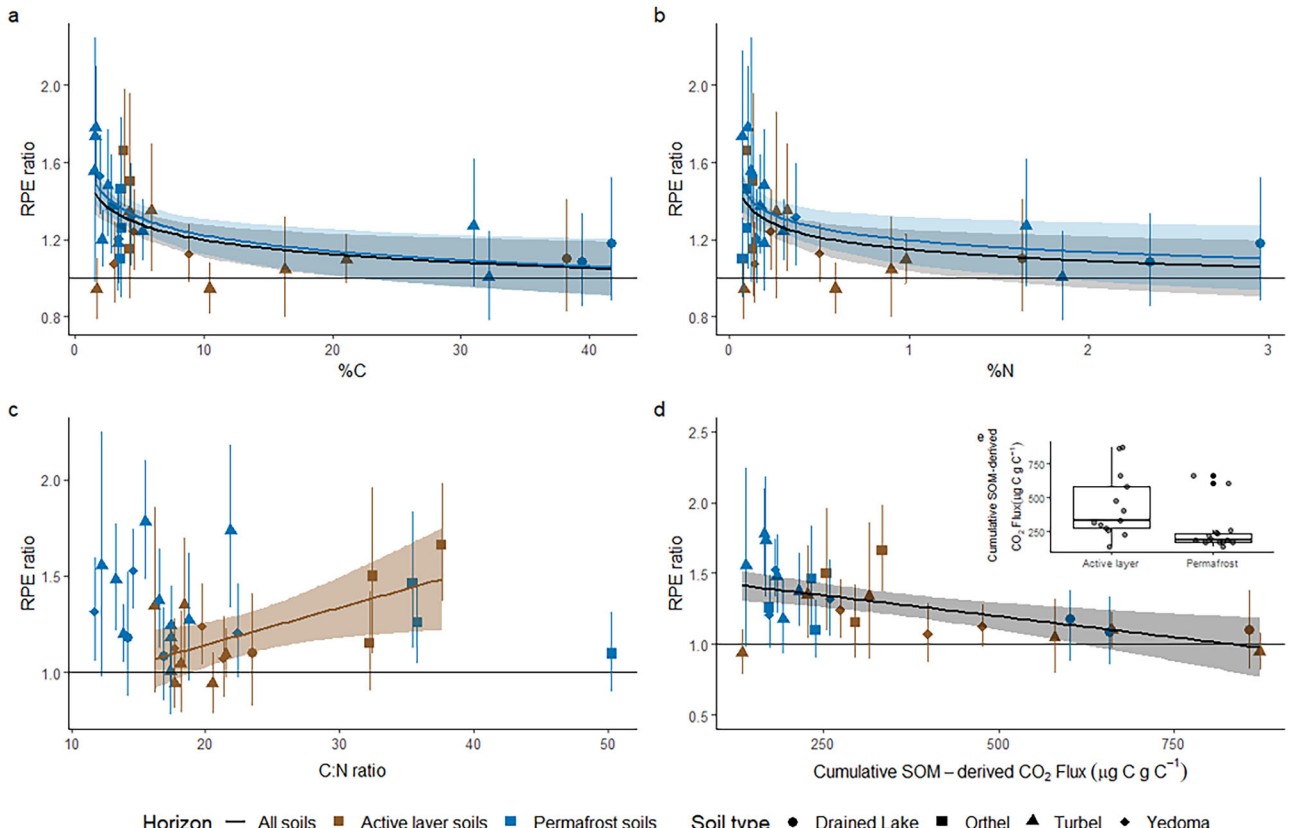

**Fig. 2 | RPE ratio interactions with soil properties.** Relationships between RPE ratio of active layer and permafrost soils and soil carbon content (**a**), nitrogen content (**b**), carbon to nitrogen ratio (**c**) and cumulative SOM-derived $CO_2$ flux (**d**) with paired comparison between active layer and permafrost soils (**e**). Error bars and ribbons represent 95% confidence intervals on mean RPE ratio for each soil from days 0 to 370. Only the fitted lines of significant ($p < 0.05$) relationships are displayed. In insert box-plot (**e**) the box represents the interquartile range (IQR), spanning from the first quartile (Q1) to the third quartile (Q3). The bold horizontal line within the box denotes the median. Whiskers extend to the smallest and largest values within 1.5 times the IQR from Q1 and Q3. Data points beyond this range are plotted as outliers.

or nutrient limitation reducing the intensity of soil priming[26] in these soils.

The RPE ratio was significantly positively related to soil C:N ratio in active layer soils ($R^2 = 0.38$, $p = 0.024$) but shows no relationship in permafrost soil ($R^2 = 0.04$, $p = 0.42$) or across all soils ($R^2 = 0.03$, $p = 0.95$). Across the full experimental duration, at a given C:N ratio, RPEs were not significantly different between active layer and permafrost soils (Fig. 2c; F = 2.93, $p = 0.10$). It has been suggested that if RPEs are driven by microbial C limitation then only soils with a C:N ratio below 20 should be susceptible to RPEs[20,26] due to the stoichiometric constraints of microbial growth. However, here we observe positive RPEs in soils with C:N ratios up to 37.6 (Fig. 2c). Positive priming in Arctic soils with high C:N ratios has previously been observed[36], although this was in high-C-content peat soils, which was not the case here. The observed positive relationship between RPEs and soil C:N ratio in active layer soils here suggests that N availability plays a role in driving RPEs in active layer soils; soils with lower relative N exhibited the strongest positive priming effects, indicating that when C is available, but N is limiting, active layer soils are most vulnerable to C loss through positive RPE. Priming of active layer soils with high C:N ratios may therefore be caused by microbial and plant competition for nutrient acquisition in soils with high C and low N contents, i.e. microbial and plant N limitation as opposed to microbial C limitation. Overall, there appears to be evidence for both microbial C and microbial or plant nutrient limitation playing key roles in controlling the magnitude of priming effects, with a potentially greater role for C limitation in some permafrost soils,

especially turbels. Detailed experiments are needed to investigate these mechanisms fully.

## Nutrient additions temporarily reduce RPEs

To further test the potential for different mechanisms driving RPEs in active layer versus permafrost layers, we supplied active layer and permafrost soils with additional nutrients (equivalent to ten times microbial biomass N) once, at the beginning of the experiment in a slow release form to maintain nutrient abundance in these soils for as long as possible through the duration of the experiment (we identified small amounts of unused fertiliser in the soils at the end of the experiment). These high levels of nutrient addition were added to enable the reduction of microbial nutrient limitation in the presence of strong plant growth and therefore plant nutrient uptake.

Overall, the magnitude of RPEs in permafrost soils was reduced in the presence of additional nutrients compared to soils without nutrient addition ($p = 0.0037$, F = 8.53, Figs. 1 and 3). This was particularly prevalent early in the experimental period. There was no significant positive priming in soils with added nutrients in the initial 100 days as soils with roots had similar SOM-derived $CO_2$ fluxes to soils without roots. The absence of a significant RPE ratio over this period (RPE ratio = 1.06, 0.76–1.37 95% CI; Fig. 3a) indicates that high nutrient levels may reduce RPEs initially. This agrees with previous work identifying lower priming intensity in permafrost soils with greater N availability or following N addition in experiments using soil incubation with $^{13}$C-labelled substrates under thaw-induced increased N availability and N addition[37]. Furthermore, relationships between RPE ratio and soil C

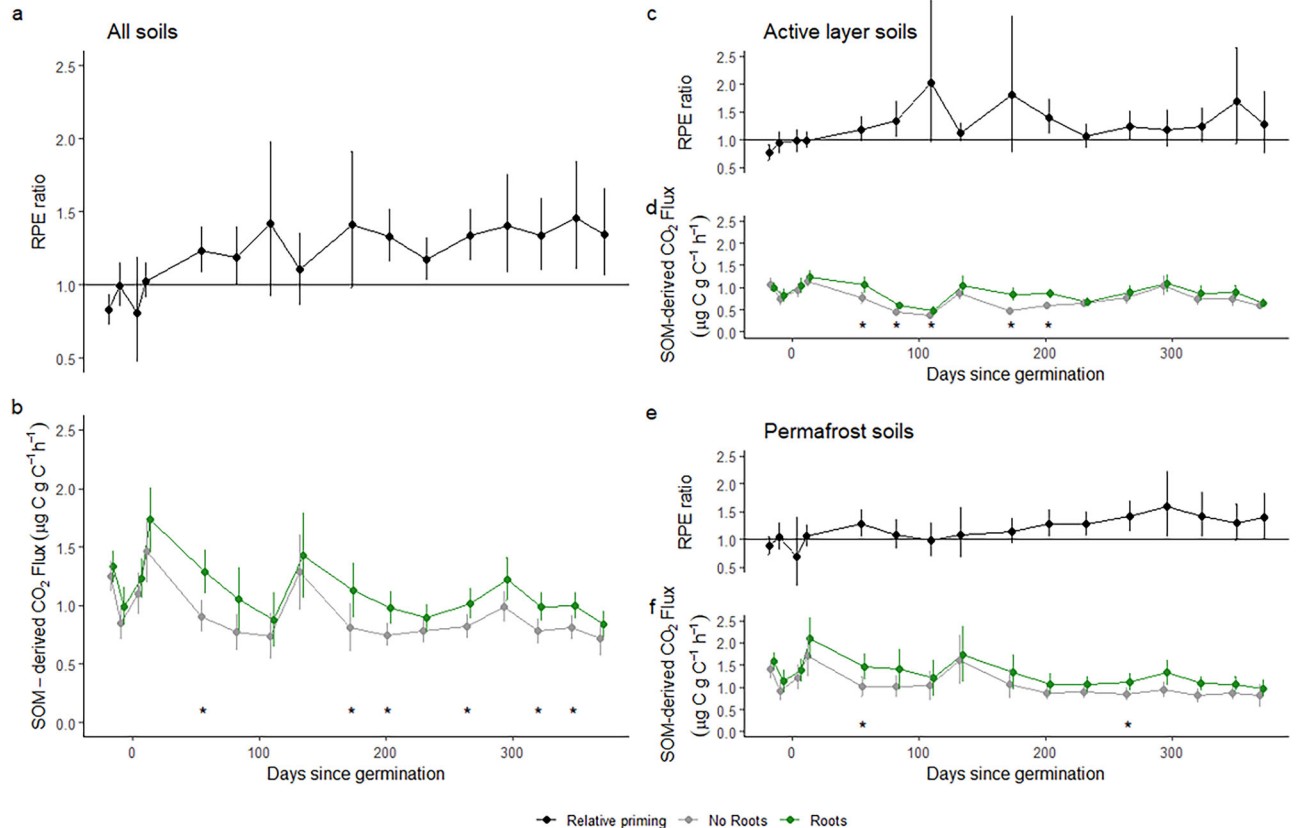

**Fig. 3 | RPE ratios and SOM-derived CO₂ fluxes from soils with nutrient addition.** RPE ratio with error bars showing 95% confidence intervals (top y-axis) and absolute fluxes of SOM-derived CO₂ with error bars showing standard error (bottom y-axis) in paired soils with and without roots in all (**a**, **b**, $n = 23$), active layer (**c**, **d**, $n = 13$) and permafrost (**e**, **f**, $n = 10$) soils with additional nutrients equivalent to ten times microbial biomass N added once at the start of the experiment. Significance differences between rooted and root-free soils, based on paired t-tests, are indicated with asterisks (*$p < 0.05$, **$p < 0.01$, ***$p < 0.001$).

content, N content and C:N ratio (Fig. 2) were lost in the presence of additional nutrients (Fig. 4). RPE ratios in soils with additional nutrients was not related to soil C content, N content or C:N ratio (prior to nutrient addition) in both active layer (C; $R^2 < 0.001$, $p = 0.97$, N; $R^2 = 0.001$, $p = 0.92$, C:N; $R^2 = 0.002$, $p = 0.89$) and permafrost (C; $R^2 = 0.14$, $p = 0.20$, N; $R^2 = 0.15$, $p = 0.19$, C:N; $R^2 = 0.12$, $p = 0.24$) soils (Fig. 4a–c). At a given C content, N content or C:N ratio, in the presences of additional nutrients RPEs were no longer significantly different between active layer and permafrost soils (C; $F = 1.56$, $p = 0.22$, N; $F = 0.74$, $p = 0.40$, C:N; $F = 0.38$). This suggests that some of the key differences between permafrost and active layer soils in terms of the magnitude of their RPEs was related to their nutrient status.

The observed reduction in RPEs in soils with nutrient addition is likely driven by an alleviation of microbial N-limitation leading to less microbial N-mining in these soils and resulting in the reduction or absence of a positive priming effect. Alternatively, the observed reduction in RPEs in soils with nutrient addition may also be driven by alleviations of plant N-limitation[38], which may have reduced resource allocation to root strategies for accessing low availability nutrient pools including exudation or organic acids and enzymes. Overall, in the permafrost and active layer soils with nutrient addition, the nutrient demand may be matched by the supply of excess nutrients, at least initially, reducing the need for N-mining and, hence, reducing positive priming effects.

Over time, it is likely that the available nutrient pool diminished, and plant nutrient demands may have started to exceed available nutrient supply in the soils. In line with such a shift; a return to positive RPEs was observed in days 101–370 (RPE ratio = 1.33, 1.25–1.41 95% CI;

Fig. 3a). This temporal effect was consistent across both active layer (RPE 1.30, 1.11–1.48 95% CI; Fig. 3c) and permafrost (RPE 1.38, 1.27–1.48 95% CI; Fig. 3e) soils, returning to a positive RPE after 185 days. It is also possible that changing soil nutrient status altered the composition and function of the microbial community[39], which may have altered the magnitude of priming effects over time. The overall reduction in the magnitude of RPEs with additional nutrients in both active layer and permafrost soils suggests that plant and microbial nutrient demand contributed to positive priming effects in both soil layers. However, based on the changes in RPEs over time in our experiment, it is likely that high levels of nutrient release associated with permafrost thaw[7,8] may merely delay the onset of positive RPEs rather than mitigate them entirely. Future studies that monitor changes in microbial community composition and function in response to changes in nutrient availability would be a valuable addition and could help to understand the relative role of microbial versus plant nutrient demand in shaping RPEs, as well as the relative roles of microbial C versus nutrient limitation.

**Implications for predicting the magnitude permafrost feedback**
Our results collectively suggest that RPEs in permafrost and active layer soils are governed by both the limited availability of high energy substrates to support microbial activity and the restricted availability of nutrients to sustain plant growth. This dual limitation highlights a complex interplay where microbial metabolism is hindered by insufficient readily decomposable organic matter, which can be alleviated by fresh C inputs from plant roots, thus activating microbes which decompose SOM. In addition, the plants also compete for nutrients,

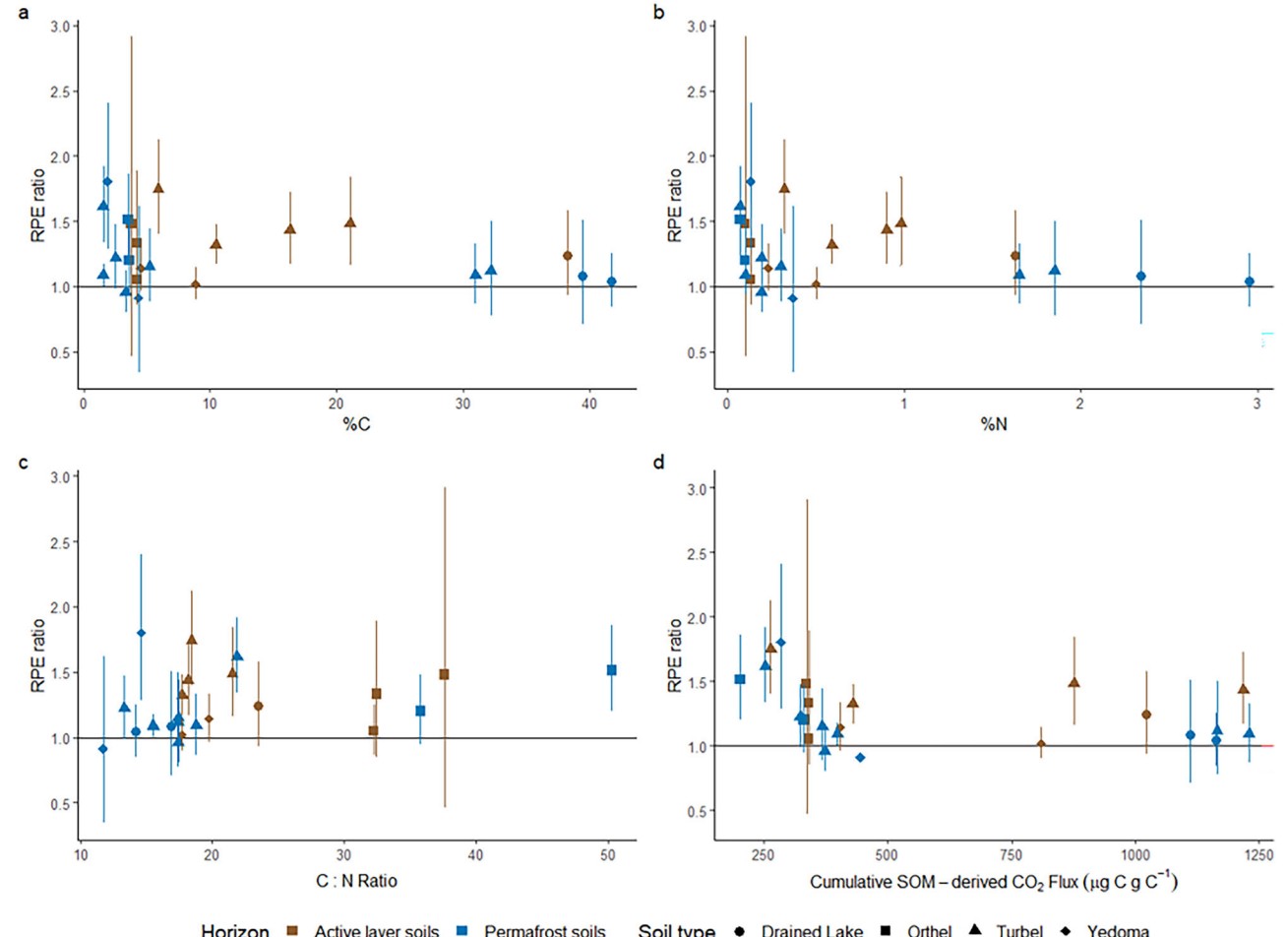

**Fig. 4 | RPE ratio interactions with soil properties from soils with nutrient additions.** Relationships between RPE ratio of active layer and permafrost soils with additional nutrients equivalent to ten times microbial biomass N and soil carbon content (**a**), nitrogen content (**b**), carbon to nitrogen ratio (**c**) and cumulative SOM-derived $CO_2$ flux (**d**). Error bars represent 95% confidence intervals on mean RPE ratio for each soil from days 0 to 370.

critical to sustaining their productivity, resulting in the breakdown of complex organic matter. The positive RPEs that we have observed thus represent interactions between plant nutrient limitation and microbial C and nutrient limitation. Further study is required to identify the relative importance of the different mechanisms in different permafrost soil types. Studies that measure the responses of microbial growth and C use efficiency to nutrient and C availability in permafrost may be critical in this area[38], with the understanding generated in combination with priming experiments likely to be crucial for predicting rates of permafrost C loss under changing climatic conditions.

In summary, we show that plant-root activity could substantially increase (by a factor of 1.31 in our study) the release of previously-frozen C following permafrost thaw. While increased carbon losses also need to be placed in the context of any changes in the rate of new SOM formation, this direct quantification of RPEs critically demonstrates that RPEs are induced by fresh C supplied by live plants. Furthermore, we have demonstrated that positive RPEs occur across contrasting soils that developed on different Quaternary superficial deposits, in different types of permafrost terrain (glaciated vs non-glaciated) and which have formed through distinct processes. Thus, RPEs are likely to increase permafrost C losses overall, and should be included in model projections of the permafrost C feedbacks and their implications for the global C budget.

## Methods

### Soil sampling

The field area selected was a transect in northwest Canada from Tuktoyaktuk on the Arctic Ocean coast through the East Channel region of the Mackenzie Delta near Inuvik, NT, across Peel Plateau, NT, to the Klondike Goldfields of interior Yukon Territory. The transect crosses from continuous permafrost near Tuktoyaktuk and in the Mackenzie Delta and Peel Plateau to extensive discontinuous permafrost in the Klondike. It also crosses from tundra in the Tuktoyaktuk Coastlands and Peel Plateau to boreal forest in the Mackenzie Delta and Klondike. The Quaternary geology and environmental context of the permafrost deposits are well known in this region[40–42]. Large vertical sections through permafrost deposits are well exposed along the coast, in lake margins, along river bluffs, in headwalls of retrogressive thaw slumps and in valley-floor gold mines, facilitating stratigraphic understanding of the soil profiles[43]. Samples were collected in August 2019 and August–September 2020 (Supplementary Table 2 & Supplementary Figs. 4–60).

Four types of permafrost soil profiles were sampled: (1) non-lake orthels, (2) drained thermokarst-lake sediments, (3) turbels, and (4) yedoma deposits (Supplementary Tables 1 and 2). The rationale was threefold: first, to compare examples from the main types of permafrost soils (Gelisols) i.e., Turbel, Orthel; second, to compare soils developed on epigenetic permafrost represented by stable till surfaces with syngenetic permafrost represented by aggraded sedimentary

deposits (yedoma silt in the Klondike and alluvial silt and sand in the Mackenzie Delta); and third, to compare soils developed in drained thermokarst-lake basins (an integral part of lowland, ice-rich permafrost terrain, which often preferentially accumulate soil organic carbon) with non-lake profiles.

The occurrence of dead roots within epigenetic permafrost is widespread along the western Arctic coast, Canada, and reflects plant growth in a deeper-than-present active layer during the early Holocene[9]. It is therefore clear that roots grew downward through newly thawed soil in a deepening active layer during the last major episode of global warming during the last glacial-to-interglacial transition prior to refreezing within aggrading permafrost in the mid to late Holocene.

All collected soil samples where frozen upon return to the field station and shipped frozen to the University of Exeter where they remained in frozen storage until processed for the beginning of the experiment.

## Soil sample analyses

Soil C and N content was quantified by running a 15 mg (+/−0.5 mg) of each homogenised and ground soil sample on a Flash 2000 Elemental Analyser (Thermo Scientific). Ethylenediaminetetraacetic acid (EDTA) was used as standards at the start of the CN run and as 'check standards' every ten samples. The EDTA standard is 9.59% N (+/−3% accepted) and 41.1% C (+/−1% accepted).

Soil primary texture analysis on all samples was measured on a Bettersizer S3+ (Bettersize Instruments Ltd., Dandong, China) soil particle analyser. Organic material was removed from 1 g of soil by digestion in 10 ml 30% hydrogen peroxide at 100 °C for 2–3 h until the supernatant was clear. Samples were then suspended in 1% sodium hexametaphosphate to aid dispersion and sonicated at low energy immediately prior to analysis.

## Mesocosm design

The mesocosms for this experiment were constructed to enable plant growth in an inert substrate in a central compartment with root access via a 2 mm nylon mesh to three soil-filled side compartments. Mesocosms were built using 5 mm thick grey PCV sheet machine cut to size and glued together using UNI-100 PVC cement (Griffon, Bolton Adhesives, Rotterdam, NL). All joints were sealed with Aqua mate, a solvent free aquarium sealant (Everbuild, Sika, UK). The whole mesocosm measured 30 cm L × 20 cm W × 20 cm H. The central compartment was 10 cm L × 2 cm W × 20 cm H. Each of the four side compartments was 10 cm L × 10 cm W × 20 cm H and filled to 10 cm with soil, creating a -1 L headspace (10 × 10 × 10 cm).

The side compartments were sealed with lids cut from 5 mm thick PVC sheet spanning two side compartments with insets of 10 ×10 cm to create a two-sided surface for the lid to seal onto the mesocosm base. An airtight seal was achieved using non-setting putty (Plumbers Mait, EVO-STIK, UK) between the mesocosm base and lid. Two holes in the mesocosm lid allowed the entry of a 6 mm O.D. inlet tube and a 9 mm I.D. outlet tube for each side compartment.

## Experimental design

A total of 54 mesocosms were used in this experiment. Thirty-one mesocosms had a temporal root exclusion mesh regime and included the broadest possible range of soil types, C and N content and soil texture. The remaining 23 mesocosms had additional nutrients added to the soils (Supplementary Fig. 61). Whilst we did not have the space to incorporate all 31 soils in the nutrient addition part of the experiment, the 23 soils selected covered the full range of soil types, C and N content and soil textures. The statistical effects of the uneven n number were checked and found to have no significant effect.

In the mesh regime mesocosms one side compartment had a 1 μm root exclusion nylon mesh (Normesh Ltd, Lancaster, UK) inserted from the beginning of the experiment (12th June 2023) and remained throughout, creating a root-free control. To investigate the relative effects of C inputs from live roots, and associated plant nutrient uptake, versus C and nutrients inputs through fresh root litter, in two of the compartments the roots were severed at different times points. To achieve this a second root-exclusion mesh was inserted into the side compartment diagonally opposite the root-free control side compartment on 14th November 2023, and a further root-exclusion mesh was inserted into the side compartment next to the root-free control side compartment on 25th March 2024. Prior to insertion of the mesh, the roots grown into the side compartment soil were severed with a knife, creating root necromass/litter. Once inserted all root-exclusion meshes remained in place until the end of the experiment 11th July 2024.

In the mesocosms with additional nutrients, one side compartment had a 1 μm root-exclusion nylon mesh (Normesh Ltd, Lancaster, UK) inserted from the beginning of the experiment (12th June 2023) and remained throughout, creating a root-free control. The remaining three side compartments had continuous root access throughout the duration of the experiment. Additional nutrients were added to these soils once at the beginning of the experiment (12th June 2023) in the form of slow release Velvit Nutrilong V180 controlled release compound fertiliser (29%N, 4%P, 8%K, 1.7%MgO, C:N = 0.51), chosen for its 6-month release period and low C content. Additionally, permafrost thaw releases a range of nutrients[44,45] which we have tried to capture with this compound input. The amount of fertiliser added to each side compartment was tailored to the soil within each mesocosm. Of the three rooted side compartments, one had no additional nutrients, one had "low" additional nutrients equivalent to 2× microbial biomass N, and one had "high" additional nutrients equivalent to 10× microbial biomass N. By making the nutrient addition proportional to microbial biomass we aimed to alleviate microbial nutrient demand. The root-free control side compartment was given the same nutrient treatment as the high additional nutrient side compartment (Supplementary Fig. 61). Despite previous studies not observing strong direct effects of nutrient additions on respiration in root-free soils[46], in our experiment, the "high" additional nutrient soils showed higher fluxes, therefore we limited our analysis to the "high" nutrients, with and without roots.

## Plant growth and maintenance

*Agrostis capillaris* was grown in the mesocosm central compartment to provide the delivery of fresh C inputs to previously frozen soils in the surrounding compartments. Although *A. capillaris* does not represent the varied vegetation types at field sampling sites (Supplementary Figs. 4–60) it was used as it combines a distribution that includes high northern latitudes with high germination rates from seed and an ability to establish and grow well in growth chamber studies. This was deemed the best combination for geographical relevance and experimental application within the physical and temporal constraints of a growth chamber experiment.

*Agrostis capillaris* seeds were sterilised using a 30% hydrogen peroxide solution. All seeds were suspended in a 30% hydrogen peroxide solution for 10 min and then rinsed thoroughly with mili-Q water. This was repeated three times before drying the seeds for 48 h at room temperature.

The central compartment of each mesocosm was filled with Field and Fairway Ceramics (Profile, Buffalo Grove, Illinois) as this inert substrate demonstrated good water retention, resistance to compaction and would encourage root growth into soil-filled side compartments. The ceramics was mixed with 4 g Velvit Nutrilong V180 controlled release compound fertiliser (29%N, 4%P, 8%K, 1.7%MgO, C:N = 0.51). This application rate was used as the ceramics contained very limited nutrients (KCl extractable; $NH_4 = 0.16 \pm 0.06$ μg gDW$^{-1}$, $NO_3 = 0.40 \pm 0.17$ μg gDW$^{-1}$, and NaHCO$_3$ extractable; $PO_4 = 0.50 \pm 0.05$ μg gDW$^{-1}$, $n = 5$), and tests with different rates of

application identified these concentrations as sufficient to promote strong *A. capillaris* germination and allow for root colonisation of side compartments. The ceramics mix was then wetted to a 1:0.65 ceramics to water ratio by weight prior to sowing. *A. capillaris* seeds were sown onto all mesocosm central compartments at a density of 1.8 mg cm$^{-2}$ on 15$^{th}$ June 2023 and germinated fully within the growth chamber by 26$^{th}$ June 2023.

The *A. capillaris* was watered twice weekly with varying quantities of deionised water ranging from 50 to 500 ml depending on plant growth stage and water demands. Similar soil moisture was maintained across mesocosms and side compartments throughout the experiment (Supplementary Fig. 2). Once fully mature the *A. capillaris* shoots where clipped once per month or when shoots reached ≥50 cm height.

### Labelling growth chamber

The experiment was set up and maintained in a bespoke, climate-controlled growth and continuous $^{13}$C-labelling chamber (BLOSOM)[29].

Permafrost soil temperature was maintained at 5 °C by partly submerging (15 cm water height) mesocosms in a large water bath cooled by a 4 kW FL4003 recirculating cooler (Julabo UK Ltd). Chamber air temperature was maintained at 20–22 °C by a bespoke closed-loop air conditioning system (Cambridge HOK, UK).

Grow lighting within the chamber was run on an inverted diurnal cycle with a 16 h photoperiod, using five BX120c2 and five BX180c2 LED strip lights (Valoya Ltd, Helsinki, Finland) set to 250–300 μmol m$^{-2}$ s$^{-1}$.

To achieve continuous labelling at ~500‰ δ$^{13}$C, compressed air (10–30 ppm), unlabelled $CO_2$ (99.8% pure, BOC, UK) and labelled $CO_2$ (99% $^{13}CO_2$, <2% $^{18}$O, CK isotopes Ltd, UK) controlled by precision mass flow controllers (Bronkhorst UK Ltd, Newmarket, UK) was sequentially mixed and supplied to the chamber. The average δ$^{13}$C isotopic composition of chamber air during the lights-on period was 506.11 ± 26.95‰ (mean ± SD).

### Flux measurements

$CO_2$ flux measurements were carried out using a bespoke push-and-pull-through respiration system[29] which continuously flowed $CO_2$-free air through side compartment headspaces at ~20 ml min$^{-1}$ controlled by analogue flow meters (Dwyer Instruments). During measurements the out-flow from each side compartment were connected to a Picarro G2201-i isotopic analyser (Picarro Inc. Santa Clara, CA) sequentially for 3–4 minutes until $CO_2$ was stable.

Flux measurements of all 216 side compartments were conducted once prior to *A. capillaris* sowing, once prior to germination and monthly for 370 days after germination. The flux of SOM-derived $CO_2$ from each side compartment was then derived by Eqs. (1) and (2).

Soil moisture measurements were carried out eight times throughout the experiment by inserting an IMKO H2 Pico32 110 mm soil moisture probe (IMKO Micromodultechnik GmbH, Germany) into the soil in each side compartment. Soil moisture calibration curves were developed for each soil independently. Reported moisture soil moisture is the average of two measurements, one at 5 cm and 10 cm soil depth.

### Statistical analyses

All statistical analyses were performed using R version 4.3.0 (2023-04-21 ucrt). Paired t-test were performed to determine the statistical significance of differences in SOM-derived $CO_2$ fluxes between paired side compartments with and without roots. Within group normality was confirmed by Shapiro-Wilk test. A repeated measures ANOVA was used to compare RPE ratios from active layer and permafrost soils over time. Normality of residuals was confirmed by Shapiro-Wilk test. Relationships between RPE ratio and measured soil elements were modelled using a nonlinear power function ($y = ax^{-b}$) for C and N content and a linear model for C:N ratio. Models were selected by comparing goodness of fit using the Akaike information criterion

(AIC). Likelihood ratio tests were performed to determine statistical significance of these models. ANCOVAs were performed to assess relative differences between active layer and permafrost soil data covarying with soil C content, N content and C:N ratio.

The concentration of SOM-derived $CO_2$ in each side compartment headspace was calculated by the following equation:

$$SOM - derived\ CO_2(ppm) = \left( \frac{\delta^{13}C_{chamber}(‰) - \delta^{13}C_{SC}(‰)}{\delta^{13}C_{chamber}(‰) - \delta^{13}C_{soil}(‰)} \right) * CO_{2_{SC}}(ppm) \quad (1)$$

Where chamber is the chamber environment, SC is the side compartment headspace and soil is the soil within the side compartment.

The flux of SOM-derived $CO_2$ from each side compartment soil was then calculated by the following equation:

$$SOM - derived\ CO_2 flux \left( \mu g\ C\ gC^{-1}h^{-1} \right)$$
$$= \left( \frac{\left( \frac{SOM-derived\ CO_2(ppm) * Flow_{air}(mlmin^{-1})}{Vm(mlmol^{-1})} \right) * M_{CO_2}}{C(g)} \right) * 60 \left( min\ h^{-1} \right) \quad (2)$$

Where Flow$_{air}$ is the incoming air flowing into the headspace, Vm is the molar volume of gas at standard temperature and pressure, M$_{CO2}$ is the molar mass of $^{12}CO_2$ and C is the mass of carbon in the soil sample derived from soil C content (%) and soil dry weight.

RPE ratio for each measured time point was then calculated by the following equation:

$$RPE\ ratio = \exp^{mean\left( \ln\left( \frac{SOM-derived\ CO_2 flux_{Roots}}{SOM-derived\ CO_2 flux_{No\ roots}} \right) \right)} \quad (3)$$

Where SOM-derived $CO_2$ Flux$_{Roots}$ is the SOM-derived $CO_2$ flux from a side compartment with roots and SOM-derived $CO_2$ Flux$_{No\ Roots}$ is the SOM-derived $CO_2$ flux from a paired side compartment without roots.

## Data availability
All data generated for this publication are or will be available from the Environmental Information Data Centre (EIDC). Carbon and nitrogen content data from soils used in this publication can be found at https://doi.org/10.5285/54e9d93f-bacb-41e4-a66a-0cdcbdd02101. All soil flux data and soil particle size data can be found at https://doi.org/10.5285/a80c688c-0a91-41ce-a8e9-89479a38a566.

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

## Acknowledgements

We thank Thomas Opel, James Pokiak and Ryan McLeod for assisting with fieldwork. Scott Dallimore (Geological Survey of Canada, Sidney, BC), The Aurora Research Institute (Inuvik, NT), Fabrice Calmels and Louis-Philippe Roy (Yukon College, Whitehorse, YT) provided logistical support. Duane Froese and Elizabeth Hall are thanked for guidance about field sites in lands of the Tr'ondek Hwech'in First Nation in the Klondike. Myles Carson kindly provided site access to Little Blanche Mine. We thank the Inuvik Hunters and Trappers Committee, the Aklavik Hunters and Trappers Committee, and the Tuktoyaktuk Hunters and Trappers Committee for approving fieldwork on Inuvialuit Private Lands. The study and all authors were funded by the Natural Environmental Research Council, grant reference N E/S010122/1. For the purpose of open access, the author has applied a 'Creative Commons Attribution (CC BY) licence to any Author Accepted Manuscript version arising from this submission.

## Author contributions

I.P.H., G.K.P., J.B.M. and G.H. conceived of the idea. J.B.M. and S.V.K. identified appropriate sampling sites and collected all the permafrost soils. N.L.F. and I.P.H. designed the experiment and experimental facility. N.L.F. carried out the experiment and analysed the data. N.L.F. wrote the manuscript with contributions from all authors.

## Competing interests

The authors declare no competing interests.

## Ethics

This research was conducted with respect for the lands and communities in which the study took place. Fieldwork was carried out in collaboration with local researchers, ensuring appropriate permissions and engagement with relevant stakeholders. We acknowledge and appreciate the support of the Tr'ondëk Hwëch'in First Nation, the Inuvik Hunters and Trappers Committee, the Aklavik Hunters and Trappers Committee, and the Tuktoyaktuk Hunters and Trappers Committee, who approved research activities on Inuvialuit Private Lands. The research team is committed to fostering an inclusive and diverse scientific environment. This study involved contributions from researchers at different career stages and disciplines, promoting collaboration across institutions. All authors contributed to the design, execution, and interpretation of the study, ensuring a fair and equitable distribution of responsibilities.
