## [Peer Review file · Nature Communications]

Positive rhizosphere priming accelerates carbon release from permafrost soils

Corresponding Author: Dr Nina Friggens

Version 0:

Reviewer comments:

Reviewer #1

(Remarks to the Author)
General comments

Friggens et al. explored an interesting topic about the rhizosphere priming effects in permafrost soils, based on mesocosm experiment where plants were grown in active layer and permafrost soils in a temperature-controlled environment with continuous ^{13}C labeling. The authors detected positive rhizosphere priming effects accelerated carbon release from both active layer and permafrost soils, and also found the rhizosphere priming effects were associated with soil carbon and nitrogen content, and also soil C:N ratio. Several key issues raised during my review on this manuscript, which I think should be addressed in any revised version before considering publication.

(1) The basic background about this study should be clearly described throughout the manuscript. Particularly, how deep of the permafrost layer at each sampling site and each sampling profile? How about the vertical distribution of root depth at each sampling site and each sampling profile? Whether could the live roots be distributed into the deep permafrost layer at each sampling site? Based on supplementary Table 2, it seemed like that some of permafrost samples were collected from very deep layers, such as 1.5-2.0 m. It is not clear whether root growth could approach this very deep soil layer even after permafrost thaw. In addition, the authors hypothesized the increased rooting depth after permafrost thaw. Is there any clear evidence about this phenomenon? Is there any evidence to show root depth could be penetrated into the very deep permafrost soils examined in this study? If not, it is less meaningful to explore rhizosphere priming effects in deep permafrost soils. These basic questions are fundamental for this study, should be fully addressed.

(2) Some of experimental conditions should be better explained and greatly improved. Based on the supplementary figures (Figure S6-58), we can see that vegetation types varied from these sampling sites from tundra, shrub to forests. However, in the mesocosm experiment, the authors only planted one species, i.e., *Agrostis capillaris*. Obviously, this situation could not reflect the in-situ conditions. Moreover, why 10 times microbial biomass N were added during the nutrient flush experiment? Why was the compound fertilizer (29%N, 4%P, 8%K, 1.7%MgO, C:N = 0.51) used to simulate nutrient release upon permafrost thaw? In addition, there are a lot of potential variables which could affect priming effects, such as soil substrate quality derived from NMR analysis, and soil physico-chemical protection et al. However, this study only measured very simple variables like soil carbon and nitrogen content, which meant too much other potential variables were not determined and ignored in this experiment. I urge the authors to take more efforts on this aspect.

(3) The detailed experimental results for each sampling site and each sampling soil types should be clearly described. Currently, the authors only showed the overall patterns of the rhizosphere priming effects for all soils and for active layer and permafrost soils. Given the large spatial heterogeneity among various sampling sites and sampling soil types, the readers are eager to know how about the rhizosphere priming effects for each sampling site and each sampling soil type. Whether did the positive rhizosphere priming effects occur across all each sampling sites and each sampling soil types?

(4) Some of results need to be better clarified, and some of discussion needs to be supported by the experimental data. For example, the authors argued that rhizosphere priming ratio in active layer soils was 1.39 in days 0-185 days and 1.03 between days 186-370. This transition date should be clearly shown in the related figure so that the readers could capture this information very quickly. Similar issues existed in other sections where the specific date was described. In addition, several discussion parts lack data support. For example, the authors argued that rhizosphere priming effects-driven carbon loss may be offset by new SOM formation, but did not provide any experimental data. Similarly, those arguments about the diminishing rhizosphere priming effects over time lack data support.

Specific comments

Line 22: How about the results for each soil type?

Line 29-30: It would be great to discuss the potential reasons for the stronger priming effects in permafrost than active layer soils in the Discussion section.

Line 41-43: Please provide the direct evidence to support this phenomenon, i.e., the increased rooting depth upon permafrost thaw.

Line 58: The authors did not measure soil substrate quality in this study.

Line 62: The sub-title should not occur among paragraphs within the Introduction section.

Line 70: Several studies derived from the Tibetan permafrost regions should not be ignored. They have clearly determined priming effects in permafrost regions.

(1) He M et al., 2023. Priming effect stimulates carbon release from thawed permafrost. *Global Change Biology*, 29: 4638-4651.

(2) Chen LY, et al., 2018. Nitrogen availability regulates topsoil carbon dynamics after permafrost thaw by altering microbial metabolic efficiency. *Nature Communications*, 9: 3951.

Line 76: Will this situation occur definitely?

Line 104-105: How about the results for each soil type and each sampling site?

Line 120-123: Do these results mean that the reduced plant vigour should be the major reason? In addition, these arguments are purely deductive, lack direct data support. Could you add more experimental data to support these arguments?

Figure 1 and 3: It would be better to clearly show the threshold date from whether the rhizosphere priming effects changed.

Line 136-137: It would be better to give more explanations for this result.

Figure 2: As mentioned in my general comments, there are a lot of potential variables which could affect priming effect. Why not consider them?

Line 174: Why 10 times microbial biomass N?

Line 186 and line 193: It would be better to clearly show the threshold date from whether the rhizosphere priming effects changed in figure 3.

Line 215-228: These arguments need data support.

Line 310-328: More details should be described about field sampling. For example, how many sampling sites or profiles were obtained during the whole survey? How many for each soil types? What about the sampling depth for active layer and permafrost? How about the active layer thickness? How about the root depth distribution for each sampling site? Whether the root distribution could approach the sampled permafrost soils across the various study sites?

Line 367-369: Why was the number of mesocosms not similar? How about the potential influence?

Line 387-389: Please give the detailed explanations why use this kind of fertilizer. Could it better characterize the nutrient release upon permafrost thaw?

Line 399: Please give the detailed explanations why use this species, given that vegetation types varied a lot among various sites.

Reviewer #2

(Remarks to the Author)

Nina L. Friggens and colleges have executed a really interesting and probably also laborious experiment to elucidate the effect of rhizodeposits on the decomposition of SOM in active-layer and permafrost soils. They find higher SOM-derived CO₂ in soils with ingrown roots than in soils without roots, and were able to attribute this increase to positive priming by ¹³C-labelling the plants and their respiration. The authors also want to shed light on the mechanism behind the priming effect by adding nutrients to the soil, thereby alleviating a possible microbial N limitation and taking away the need for microbes to mine the SOM. Their research question is relevant and the experiment is thoughtful and looks well-executed.

I feel the core of the experiment, i.e., the assessment of the rhizosphere priming effect in mesocosms, has been executed well and the experiment is well-thought of. However, I have trouble finding the rationale behind the addition of a plant fertiliser (Velvit Nutrilong V180) to a soil which is probably already high in nutrients because of thawing-induced SOM

mineralisation and then attributing the changes in priming to the alleviation of microbial N-limitation. The nutrients in the soil are not measured afterwards, nor is the microbial community characterised or microbial biomass measured, which are all key elements possibly supporting or debunking the hypotheses stated in this paper.

An overall weak point of the MS is the superficial interpretation and discussion of the results. For example, the authors do not elaborate on the distinction between microbial and plant N-limitation possibly steering the observed dynamics and thus being crucial for the interpretation of the results. Also they mention 'RPEs being driven by microbial C limitation' (r153), but do not further elaborate on this finding and only discuss on N limitation further on. The short digression on SOM formation because of higher plant productivity should be left out in favour of a more thorough discussion and literature check of all findings.

r49/71/90: Please use rhizodeposits/rhizodeposition vs root-exudates/root-exudation consistently throughout the MS.

r84-85: I don't understand this sentence. How can plant-derived C be partitioned in two types of soil C?

r103: I find the term unlabelled CO₂ confusing. Without thorough reading of MM before this stage stating that plants are grown entirely on labelled CO₂, it could also mean root respiration or something else. I suggest to clarify this term before, or use something more specific such as SOM-derived CO₂.

r108: Supplementary fig. 2 is not referred to anywhere in the MS.

r108-109: I'm struggling with the time scales of this experiment. Root exudates are labile C-components, so when root exudates are halted because of severing plant roots, you would expect a much faster reaction of microbial respiration than 'within a month', even if soil temperature was only 5°C in your experiment. See eg. (Müller et al., 2022), where these kind of dynamics change overnight. Why did you only measure CO₂ concentration a month after severing roots?

Figure 1&3: I'm not too fond of two-axes figures because they are confusing and unnecessary; the scales of the axes are different and the units are not comparable, standard error bars and 95% confidence bars are used in the same window. But of course, this is more of a formatting issue.

Figure 2: The relationship between %C, %N and RPE ratio looks hardly linear. It seems that RPE ratio is higher when %C < 5 and %N < .5 or so, and then there is a flat line when %C is between 5 & 40%, and N is between .5 and 3%. Given the much larger range of the flat line and the fact that you are also comparing active layer and permafrost soils, this is not trivial. A boxplot showing %C and %N categories, and analysis with an ANOVA would statistically be more sound.

r164: if p=0.24, how is this statistically significant?

r188-189: 'may alleviate RPEs': What do you mean with this? One could understand 'may alleviate microbial N-limitation, leading to lower N-mining in SOM and resulting in the absence of a priming effect'. However, more likely than a microbial N limitation, especially in soils that had a thaw-related nutrient flush, is a plant N limitation and a microbial C limitation, which is explained nicely in this paper (Soong et al., 2020). This assumption would lead to an entirely different interpretation of the results. Moreover, no evidence or measurements are given for the statement that the available nutrient pool diminished after 185 days. The initial absence of RPE and recovery later on might be induced by other dynamics as well, like a N-induced shift in microbial community rather than RPEs being driven by nutrient demand.

r215-228: It is true that little is known about SOM formation and stabilisation in high-latitude areas, but this is not what your experiment or research is about. Maybe the MS would benefit more from a deeper interpretation of your (interesting!) results than from this digression.

r459-469: Did you test your data for within-group normality (t-test) or normality of the residuals (ANOVA). It's not mentioned here, but remains an important assumption.

r475: Please have a look at the notation of this equation. Mention units consistently (60 min h⁻¹), mend spaces (mmol⁻¹), write physical quantities and units the same for each term (g C), and take in account conversions (g → μg)

r476: Why FLOWair is capitalised when it's not in the equation?

r480: I don't understand why the RPE ratio is noted like $\exp(\ln(\text{CO}_2\text{fluxroots}/\text{CO}_2\text{fluxnoroots}))$, which just comes down to $\text{CO}_2\text{fluxroots}/\text{CO}_2\text{fluxnoroots}$ for values other than 0.

Supplementary

r24: moth → month

Müller, R., Maier, A., Inselsbacher, E., Peticzka, R., Wang, G., & Glatzel, S. (2022). ¹³C-Labeled Artificial Root Exudates Are Immediately Respired in a Peat Mesocosm Study. *Diversity*, 14(9), Article 9. <https://doi.org/10.3390/d14090735>

Soong, J. L., Fuchslueger, L., Marañon-Jimenez, S., Torn, M. S., Janssens, I. A., Penuelas, J., & Richter, A. (2020). Microbial

carbon limitation: The need for integrating microorganisms into our understanding of ecosystem carbon cycling. *Global Change Biology*, 26(4), 1953–1961. <https://doi.org/10.1111/gcb.14962>

Version 1:

Reviewer comments:

Reviewer #1

(Remarks to the Author)

I appreciate the authors for their great efforts to address my previous comments. Compared with earlier studies available in permafrost regions, this study added valuable insights into priming effects by conducting a long-term labeling mesocosm experiment (with real plants, climate-controlled growth and continuous C13 labeling chamber experiment). Several remaining minor points need to be addressed.

- (1) Line 101: “Results” section should be replaced with “Results and discussion”, since the authors also discussed the related results in this section.
- (2) Line 110: To improve the readability, it would be better to briefly mention the labeling mesocosm experiment before the detailed description of the results, given the detailed methods occurred at the end of the paper.
- (3) Figure 1 and the corresponding results section: Based on this figure, we can see that the priming effect was significant in only two incubation days for the active layer (Fig.1c). Similarly, the priming effect was not significant for all of incubation days in the permafrost (Fig.1c). The authors needed to clarify this point and discuss more for this issue.
- (4) Line 170: This argument could not be obtained directly from Fig.2d. It would be better to add an inserted panel for showing this comparison between permafrost and active layer.
- (5) Line 235: the reduction of microbial nutrient limitation.
- (6) Line 247-248: Could the authors conduct some kind of statistical comparison to illustrate this point in a more direct manner?

Reviewer #2

(Remarks to the Author)

Nina L. Friggens and colleagues have edited their manuscript thoroughly and made an effort to respond to my comments. Most of my questions or comments have been well addressed, although some experimental issues cannot be dealt with without repeating the experiment, such as the low temporal resolution of the sampling, or the lack of destructive sampling to get more insight into soil nutrient dynamics, microbial biomass or community. Although I still consider this to be a weakness of the manuscript, which inevitably leads to speculation in the discussion of plant or microbial nutrient status and what exactly causes the RPE, I agree with the authors in their reasoning that the lack of this information does not necessarily weaken their main conclusion. They have also added, as I requested, a better contextualisation of their results in the discussion, and they also now have a very good, relevant and important conclusion.

r307: Here you mention N-mining as a plant-driven process, but it's a microbial process. Also, microbial N-mining causing positive priming (r54: 'This implies that RPEs are controlled by microbial nutrient demand') is controversial and has been challenged in the literature, so this statement should be nuanced and positive priming should not be attributed to N-mining without reservation (Verbrigghe et al., 2022; Wild et al., 2019).

Verbrigghe, N., Meeran, K., Bahn, M., Fuchslueger, L., Janssens, I. A., Richter, A., Sigurdsson, B. D., Soong, J. L., & Vicca, S. (2022). Negative priming of soil organic matter following long-term in situ warming of sub-arctic soils. *Geoderma*, 410, 115652. <https://doi.org/10.1016/j.geoderma.2021.115652>

Wild, B., Li, J., Pihlblad, J., Bengtson, P., & Rütting, T. (2019). Decoupling of priming and microbial N mining during a short-term soil incubation. *Soil Biology and Biochemistry*, 129, 71–79. <https://doi.org/10.1016/j.soilbio.2018.11.014>

Reviewer #1 (Remarks to the Author):

General comments

Friggens et al. explored an interesting topic about the rhizosphere priming effects in permafrost soils, based on mesocosm experiment where plants were grown in active layer and permafrost soils in a temperature-controlled environment with continuous 3CO_2 labeling. The authors detected positive rhizosphere priming effects accelerated carbon release from both active layer and permafrost soils, and also found the rhizosphere priming effects were associated with soil carbon and nitrogen content, and also soil C:N ratio. Several key issues raised during my review on this manuscript, which I think should be addressed in any revised version before considering publication.

(1) The basic background about this study should be clearly described throughout the manuscript. Particularly, how deep of the permafrost layer at each sampling site and each sampling profile? How about the vertical distribution of root depth at each sampling site and each sampling profile? Whether could the live roots be distributed into the deep permafrost layer at each sampling site? Based on supplementary Table 2, it seemed like that some of permafrost samples were collected from very deep layers, such as 1.5-2.0 m. It is not clear whether root growth could approach this very deep soil layer even after permafrost thaw. In addition, the authors hypothesized the increased rooting depth after permafrost thaw. Is there any clear evidence about this phenomenon? Is there any evidence to show root depth could be penetrated into the very deep permafrost soils examined in this study? If not, it is less meaningful to explore rhizosphere priming effects in deep permafrost soils. These basic questions are fundamental for this study, should be fully addressed.

We thank the reviewer for highlighting that more detail was required here. To address the reviewer's concerns we have added text detailing the observed presence of roots and how this indicates downward root growth in response to active-layer deepening in the introduction (L55-60) and in greater detail in the soil sampling part of the methods section (L473-478). Additionally, we have added active-layer thickness information to supplementary table 2. Additionally, please find a detailed response to the reviewer's specific questions below:

The occurrence of dead roots within epigenetic permafrost is widespread along the western Arctic coast, Canada, and reflects plant growth in a deeper-than-present active layer during the early Holocene (Burn, 1997). Later active-layer thinning froze the roots within permafrost. The base of the deepened active layer is marked by a thaw unconformity in the permafrost stratigraphy. Radiocarbon dating of roots and rhizomes in the basal part of the palaeo-active layer indicates that the maximum development of the active layer occurred about 11,000–9,000 calendar years ago (Burn, 1997; Murton et al., 1998). ALT in the Tuktoyaktuk Coastlands, where profiles P1 to P6 were sampled, at that time was about 2.5 times thicker than present largely because the warmer climate allowed development of forest, which trapped snow and increased ground temperatures in winter (Burn, 1997). Overall, therefore it is clear that roots grew downward through newly thawed soil in a deepening active layer during the last major episode of global warming during the last glacial-to-interglacial transition prior to refreezing within aggrading permafrost in the mid to late Holocene. Finally, while setting up the experiment we found that, while roots were more abundant in active layer samples, we can confirm root presence in all permafrost soils.

Burn CR. 1997. Cryostratigraphy, paleogeography, and climate change during the early Holocene warm interval, western Arctic coast, Canada. *Canadian Journal of Earth Sciences* 34, 912–925.

Murton JB, French HM, Lamothe M. 1998. The dating of thermokarst terrain, Pleistocene Mackenzie Delta, Canada. In: Lewkowicz AG, Allard M. (eds) *Proceedings of the 7th International Permafrost Conference*, Yellowknife, Canada, 23–27 June 1998, Collection Nordicana, Centre d'études nordiques, Université Laval, pp. 777–782.

Murton JB, Goslar T, Edwards ME, Bateman MD, Danilov PP, Savvinov GN, Gubin SV, Ghaleb B, Haile J, Kanevskiy M, Lozhkin AV, Lupachev AV, Murton DK, Shur Y, Tikhonov A, Vasil'chuk AC, Vasil'chuk YK, Wolfe SA. 2015. Palaeoenvironmental interpretation of yedoma silt (ice complex) deposition as cold-climate loess, Duvanny Yar, northeast Siberia. *Permafrost and Periglacial Processes* 26, 207–288.

(2) Some of experimental conditions should be better explained and greatly improved. Based on the supplementary figures (Figure S6-58), we can see that vegetation types varied from these sampling sites from tundra, shrub to forests. However, in the mesocosm experiment, the authors only planted one species, i.e., *Agrostis capillaris*. Obviously, this situation could not reflect the in-situ conditions. Moreover, why 10 times microbial biomass N were added during the nutrient flush experiment? Why was the compound fertilizer (29%N, 4%P, 8%K, 1.7%MgO, C:N = 0.51) used to simulate nutrient release upon permafrost thaw? In addition, there are a lot of potential variables which could affect priming effects, such as soil substrate quality derived from NMR analysis, and soil physico-chemical protection et al. However, this study only measured very simple variables like soil carbon and nitrogen content, which meant too much other potential variables were not determined and ignored in this experiment. I urge the authors to take more efforts on this aspect.

These are all fair points, and we are glad to have to opportunity to clarify. It is highly challenging/impossible to simulate field conditions fully in the laboratory, and we believe our approach is the correct balance between realism and what is experimentally possible, and goes substantially beyond previous studies in this regard. Firstly, it would be very difficult to recreate tundra and northern boreal plant communities in the control growth chamber system at the scales used. Therefore, *Agrostis capillaris*. was considered the ideal plant species as it combines a distribution that includes high northern latitudes with high germination rates from seed and an ability to establish and grow well in growth chamber studies. Text further explaining this has been added to the manuscript (L554-556).

Ten times microbial biomass N were added to soils to simulate a thaw-related nutrient flush. This level of nutrient addition was chosen to experimentally push the soil system into a state of nutrient abundance which would last for an extended time period even in the presence of a large nutrient sink namely the actively growing plants. This is also why a compound fertiliser specifically designed to release nutrients over a long period of time was used. The use of a fertiliser containing multiple macro nutrients was deemed to more accurately reflect the range of nutrients released as a result of permafrost thaw (Finger et al. 2016; Yang et al. 2021), as opposed to a pure N addition often used in these types of experiments. Text further explaining this has been added to the manuscript (L301-305 & L539-541).

It is true that many potential variables may influence soil priming effects, and it will be interesting to explore in future work building on the current study. Our approach to identifying controls on priming effects is consistent with other high-latitude and global synthesis studies which have considered organic matter contents and C:N ratios as dominant drivers of decomposition rates (e.g. Schadel et al GCB). We have included the major permafrost soil types represent in this region. To highlight this further to the reader we have added 'Soil type' as a shape variable to Figures 2 & 4 in the manuscript. In addition to this we have expanded Supplementary Figure 1 to include plots examining the relationship between RPEs and

cumulative SOM-derived flux (as a proxy for SOM bioavailability) and primary texture variables. We also identify priorities for future work that reflect the very valid suggestions from this reviewer.

Finger RA, Turetsky MR, Kielland K, et al (2016) Effects of permafrost thaw on nitrogen availability and plant–soil interactions in a boreal Alaskan lowland. *Journal of Ecology* 104:1542–1554. <https://doi.org/10.1111/1365-2745.12639>

Yang G, Peng Y, Abbott BW, et al (2021) Phosphorus rather than nitrogen regulates ecosystem carbon dynamics after permafrost thaw. *Global Change Biology* 27:5818–5830. <https://doi.org/10.1111/gcb.15845>

(3) The detailed experimental results for each sampling site and each sampling soil types should be clearly described. Currently, the authors only showed the overall patterns of the rhizosphere priming effects for all soils and for active layer and permafrost soils. Given the large spatial heterogeneity among various sampling sites and sampling soil types, the readers are eager to know how about the rhizosphere priming effects for each sampling site and each sampling soil type. Whether did the positive rhizosphere priming effects occur across all each sampling sites and each sampling soil types?

Thanks for this suggestion. In figures 2 and 4 in the manuscript we have included information on RPE ratio split by soil type denoted by the shape of the points in these plots. This will help the reader see that there are no overall patterns in RPE ratios by soil type both with and without nutrient addition. We are unable to explore relationships between RPE ratios and sampling sites as our experimental design only replicates soil types, not sampling sites.

(4) Some of results need to be better clarified, and some of discussion needs to be supported by the experimental data. For example, the authors argued that rhizosphere priming ratio in active layer soils was 1.39 in days 0-185 days and 1.03 between days 186-370. This transition date should be clearly shown in the related figure so that the readers could capture this information very quickly. Similar issues existed in other sections where the specific date was described. In addition, several discussion parts lack data support. For example, the authors argued that rhizosphere priming effects-driven carbon loss may be offset by new SOM formation, but did not provide any experimental data. Similarly, those arguments about the diminishing rhizosphere priming effects over time lack data support.

As requested, we have generated the plot with edits which denote the durations over which we summarise “priming” and “no priming” using variations in the plot background colour. We have included the modified plots below.

Whilst we see the reviewer’s point about enabling the reader to capture information quickly, we feel that the requested edits to the plots only make them busier and harder to interpret at a glance, but if the reviewer disagrees, we can update the graphs in the manuscript to the ones included below. To improve the readability of Figures 1 & 3 in the manuscript (and in response to a suggestion from Reviewer 2) we have simplified the plots, removed the secondary y-axis and plotted RPE ratio and SOM-derived CO₂ flux separately for all soils and each horizon.

In response to a comment by reviewer 2 the section on the potential for RPE-driven carbon loss to be offset by new SOM formation has been removed in order to place more emphasis on experimental data.

Specific comments

Line 22: How about the results for each soil type?

In figures 2 and 4 in the manuscript we have included information on RPE ratio split by soil type denoted by the shape of the points in these plots. This will help the reader see that there are no overall patterns in RPE ratios by soil type both with and without nutrient addition.

Line 29-30: It would be great to discuss the potential reasons for the stronger priming effects in permafrost than active layer soils in the Discussion section.

We have added additional text further discussing the stronger priming effects in permafrost than active layer soils (L200-238). Reviewer 2 has requested more detailed and in-depth discussion of the experimental results presented here. In order to address both reviewers' requests we have added additional detail and discussion of results throughout the manuscript. We do not include a specific discussion section (as permitted by the journal formatting guidelines), as we feel the results lend themselves to be discussed as they arise in this case.

Line 41-43: Please provide the direct evidence to support this phenomenon, i.e., the increased rooting depth upon permafrost thaw.

Additional text has been added to the introduction (L55-60) and soil sampling part of the methods section (L473-478) to provide additional support for the statements regarding increased rooting depth upon permafrost thaw. See also response to general comment 1 above.

Line 58: The authors did not measure soil substrate quality in this study.

The reviewer is correct that substrate quality was not measured in this study. However, the point we make in the introduction is a theoretical one highlighting that SOM quality may affect soil priming and we therefore feel it remains relevant to the overall point being made.

In addition to this we have expanded Figure 2 to include plots examining the relationship between RPEs and cumulative SOM-derived CO₂ flux as a proxy for SOM bioavailability or SOM quality. This is discussed in the manuscript L210-212.

Line 62: The sub-title should not occur among paragraphs within the Introduction section.

Subheadings have been removed from the introduction as per the formatting requirements.

Line 70: Several studies derived from the Tibetan permafrost regions should not be ignored. They have clearly determined priming effects in permafrost regions.

(1) He M et al., 2023. Priming effect stimulates carbon release from thawed permafrost. *Global Change Biology*, 29: 4638-4651.

(2) Chen LY, et al., 2018. Nitrogen availability regulates topsoil carbon dynamics after permafrost thaw by altering microbial metabolic efficiency. *Nature Communications*, 9: 3951.

We thank the reviewer for the suggested literature. We have read and cited these where relevant (L216-218, L253-257, L275-277, L322-326).

Line 76: Will this situation occur definitely?

Additional text has been added to the introduction (L55-60) and soil sampling part of the methods section (L473-478) to provide additional support for the statements regarding increased rooting depth upon permafrost thaw. See also response to general comment 1 above.

Line 104-105: How about the results for each soil type and each sampling site?

In figures 2 and 4 in the manuscript we have included information on RPE ratio split by soil type denoted by the shape of the points in these plots. This will help the reader see that there are no overall patterns in RPE ratios by soil type both with and without nutrient addition. We are unable to explore relationships between RPE ratios and sampling sites as our experimental design only replicates soil types, not sampling sites.

Line 120-123: Do these results mean that the reduced plant vigour should be the major reason? In addition, these arguments are purely deductive, lack direct data support. Could you add more experimental data to support these arguments?

Even though above-ground growth rates declined and the plants required less frequent clipping as the experiment went on, the headspace CO₂ concentration remained similar throughout the experiment (Supplementary Fig. 3a) and did not decline over time, suggesting that similar levels of root activity were maintained throughout the duration of the experiment. We have edited the text to better explaining this (L174-180).

Figure 1 and 3: It would be better to clearly show the threshold date from whether the rhizosphere priming effects changed.

Please see the response to general comment 4 where we show plots with the change in priming effects denoted.

Line 136-137: It would be better to give more explanations for this result.

We have added additional text further discussing the stronger priming effects in permafrost than active-layer soils (L201-234).

Figure 2: As mentioned in my general comments, there are a lot of potential variables which could affect priming effect. Why not consider them?

The reviewer makes a good point about the many potential variables which may influence soil priming effects and it will be interesting to explore in future work building on the current study. In the current work we sought to test hypotheses linked to nutrient mining which are hypothesised to be particularly important for high-latitude soils. To address the points made by the reviewer here and in the general comments, we have expanded Figure 2 to include plots examining the relationship between RPEs and cumulative SOM-derived CO₂ flux (as a proxy for SOM bioavailability) and Supplementary Figure 1 has been expanded to include relationships between RPEs and primary texture variables.

Line 174: Why 10 times microbial biomass N?

Ten times microbial biomass N was added to soils to simulate a large level of thaw-related nutrient flush, and to reduce or eliminate microbial nutrient limitation. This is why the level of application was made proportional to microbial biomass N and was chosen to experimentally push the soil system into a state of nutrient abundance which would persist for an extended duration. This is also why a compound fertiliser specifically designed to release nutrients over a long period of time was used. Text further explaining this has been added to the manuscript (L302-306 and L545-546).

Line 186 and line 193: It would be better to clearly show the threshold date from whether the rhizosphere priming effects changed in figure 3.

Please see the response to general comment 4 where we show plots with the change in priming effects denoted.

Line 215-228: These arguments need data support.

In response to a comment by reviewer 2 the section on the potential for RPE-driven carbon loss to be offset by new SOM formation has been removed in order to place more emphasis on experimental data.

Line 310-328: More details should be described about field sampling. For example, how many sampling sites or profiles were obtained during the whole survey? How many for each soil types? What about the sampling depth for active layer and permafrost? How about the active layer thickness? How about the root depth distribution for each sampling site? Whether the root distribution could approach the sampled permafrost soils across the various study sites?

The field sampling programme collected soil samples from vertical profiles at 14 sites. Limitations in sample size and available space in the growth chamber prevented samples from two Orthels beside the East Channel of the Mackenzie River from being analysed.

The field sampling programme and the presence of roots in the sampled profiles has been described in greater detail in the soil sampling part of the methods section (L4551-478). Additionally, we have added further active-layer thickness information to supplementary table 2.

Line 367-369: Why was the number of mesocosms not similar? How about the potential influence?

In the design, we maximised the number of independent soil types that we studied, providing confidence in the priming results across contrasting permafrost soils. The nutrient manipulation experiment was included to add further mechanistic understanding, and we measured as many soil types as possible within size constraints imposed by the labelling chamber. We do not make direct statistical comparisons between these two experiments. We chose to carefully distribute the available space and soil samples in a way which enabled the greatest range of soil types, horizons, C and N contents to be analysed within the physical limitations of our experimental set up and available soils.

The statistical effects of the uneven number were checked and found to have no significant effect. Text has been added to the methods section (L519-520) to make this more explicit for the reader.

Line 387-389: Please give the detailed explanations why use this kind of fertilizer. Could it better characterize the nutrient release upon permafrost thaw?

The use of a fertiliser containing multiple macro nutrients was deemed to more accurately reflect the effects of a thaw-induced nutrient flush (Finger et al. 2016; Yang et al. 2021), as opposed to a pure N addition often used in these types of experiments. A compound fertiliser specifically designed to release nutrients over a long period of time was used so that the fertilisation effect would last for the duration of the experiment. Text further explaining this has been added to the manuscript (L301-305 & L539-541). See also response to general comment 2.

Line 399: Please give the detailed explanations why use this species, given that vegetation types varied a lot among various sites.

Agrostis capillaris was used because it combines a distribution that includes high northern latitudes with high germination rates from seed and an ability to establish and grow well in growth chamber studies. Therefore, this species is relevant to the ecosystems involved and experimentally tractable. Including the varied vegetation types is not possible given the size of the bespoke growth chamber to house trees and shrubs. Therefore, the pragmatic approach was to use a single, widespread species that was experimentally tractable. We argue that our approach still represents a major advance over priming studies that use ¹³C-labelled substrates without live plants, while also recognising that no lab experiment can fully replicate field conditions. Text further explaining this has been added to the manuscript (L554-560). Future studies that investigate the extent to which plant species or functional type affect priming rates in permafrost soils would be very valuable but were beyond the scope of this study.

Reviewer #2 (Remarks to the Author):

Nina L. Friggens and colleagues have executed a really interesting and probably also laborious experiment to elucidate the effect of rhizodeposits on the decomposition of SOM in active-layer and permafrost soils. They find higher SOM-derived CO₂ in soils with ingrown roots than in soils without roots, and were able to attribute this increase to positive priming by ¹³C-labelling the plants and their respiration. The authors also want to shed light on the mechanism behind the priming effect by adding nutrients to the soil, thereby alleviating a possible microbial N limitation and taking away the need for microbes to mine the SOM. Their research question is relevant and the experiment is thoughtful and looks well-executed.

I feel the core of the experiment, i.e., the assessment of the rhizosphere priming effect in mesocosms, has been executed well and the experiment is well-thought of. However, I have trouble finding the rationale behind the addition of a plant fertiliser (Velvit Nutrilong V180) to a soil which is probably already high in nutrients because of thawing-induced SOM mineralisation and then attributing the changes in priming to the alleviation of microbial N-limitation. The nutrients in the soil are not measured afterwards, nor is the microbial community characterised or microbial biomass measured, which are all key elements possibly supporting or debunking the hypotheses stated in this paper.

An overall weak point of the MS is the superficial interpretation and discussion of the results. For example, the authors do not elaborate on the distinction between microbial and plant N-limitation possibly steering the observed dynamics and thus being crucial for the

interpretation of the results. Also they mention 'RPEs being driven by microbial C limitation' (r153), but do not further elaborate on this finding and only discuss on N limitation further on. The short digression on SOM formation because of higher plant productivity should be left out in favour of a more thorough discussion and literature check of all findings.

We thank the reviewer for a constructive and helpful review which has greatly improved the manuscript. Furthermore, we appreciate that whilst the reviewer is rightly critical of some elements of the experiment and manuscript, they acknowledge how much work has gone into executing this piece of experimental work.

The rationale to use high levels of a slow-release compound fertiliser was threefold: a) to push the soil microbial community into a state of nutrient abundance even in the presence of a large nutrient sink namely the actively growing plants, thus removing the microbial nutrient limitation to aid testing our microbial nutrient demand hypothesis for RPEs, b) to introduce a range of nutrients as opposed to simply N, and c) to enable the slow and even release of nutrients throughout the long-term experiment. We have added text to the manuscript explaining this (L301-306, L539-541 & L545-546).

The reviewer makes a good point about measuring the soil nutrients, microbial community and biomass; this would likely have improved the strength of the deductions made about microbial nutrient limitations. However, it would have required continuous destructive sampling. While there is a flush of nutrients on thaw the strong plant growth and plant nutrient uptake necessitates the need for additional nutrient release to reduce microbial nutrient limitation. We take the reviewer's point that we have contextualised the discussion around microbial nutrient limitation, but now also discuss the results from the perspective of plant nutrient limitation as well (L280-287), recognising we cannot necessarily distinguish between them.

Throughout the manuscript we have expanded the discussion of the results presented, adding more details, thorough interrogation, discussion and contextualisation of observed patterns in our data. We believe that this has greatly improved the manuscript, and thank the reviewer again for their suggestions.

r49/71/90: Please use rhizodeposits/rhizodeposition vs root-exudates/root-exudation consistently throughout the MS.

All changed to consistently be termed 'root exudates/exudation' in text.

r84-85: I don't understand this sentence. How can plant-derived C be partitioned in two types of soil C?

Here we mean that soil C fluxes can be partitioned into plant-derived C and RPE-induced release of soil C. We have clarified the sentence in text to:

"The use of $^{13}\text{CO}_2$ and soils with and without roots allows partitioning of C effluxes from soil into i) plant-derived C and ii) RPE-induced release of soil C." (L112-114)

r103: I find the term unlabelled CO₂ confusing. Without thorough reading of MM before this stage stating that plants are grown entirely on labelled CO₂, it could also mean root respiration or something else. I suggest to clarify this term before, or use something more specific such as SOM-derived CO₂.

We like the reviewer's suggestion of using the term 'SOM-derived CO₂' in place of 'Unlabelled CO₂' as this is much clearer. We have changed the term in all relevant places in the manuscript text and figures.

r108: Supplementary fig. 2 is not referred to anywhere in the MS.

We have now added text including reference to Supplementary figure 2 to the methods section of the paper (L578-580).

r108-109: I'm struggling with the time scales of this experiment. Root exudates are labile C-components, so when root exudates are halted because of severing plant roots, you would expect a much faster reaction of microbial respiration than 'within a month', even if soil temperature was only 5°C in your experiment. See eg. (Müller et al., 2022), where these kind of dynamics change overnight. Why did you only measure CO₂ concentration a month after severing roots?

The reviewer is correct in pointing out that the drop in SOM-derived CO₂ flux likely occurred sooner than one month as root exudates are turned over and respired within days, therefore the effect on SOM-derived CO₂ fluxes is likely short lived. We were unable to capture this empirically in this experiment as fluxes were monitored on average once per month, which still represented a considerable amount of work as the reviewer highlights. It would have been valuable data to have; however, without them we can only say for certain when the drop in SOM-derived CO₂ flux occurred within the month. We would argue that this does not distract from any of our evidence of priming, we just don't have the temporal resolution to detect this rapid change. Text explaining this has been added to the manuscript (L150-155).

Figure 1&3: I'm not too fond of two-axes figures because they are confusing and unnecessary; the scales of the axes are different and the units are not comparable, standard error bars and 95% confidence bars are used in the same window. But of course, this is more of a formatting issue.

We have split the double axes plots in question into two separate single axes plots each but maintained the layout and common x-axis so that the RPE ratio and SOM-derived CO₂ fluxes can be interpreted on the same x-axis (Figs 1 & 3).

Figure 2: The relationship between %C, %N and RPE ratio looks hardly linear. It seems that RPE ratio is higher when %C < 5 and %N < .5 or so, and then there is a flat line when %C is between 5 & 40%, and N is between .5 and 3%. Given the much larger range of the flat line and the fact that you are also comparing active layer and permafrost soils, this is not trivial. A boxplot showing %C and %N categories, and analysis with an ANOVA would statistically be more sound.

We thank the review for the suggestion to look at the relationship between %C, %N and RPE ratio and consider whether linear models are appropriate. Upon re-examining these relationships using non-linear models we have found that a power function model is the best fit for the %C and %N data, while a linear model remains the best fit for the C:N data. These models were selected by comparing goodness of fit using the Akaike information criterion (AIC). We have updated all relevant plots and text in both the results (L246-253) and methods (L622-625) sections to reflect this.

We feel that using non-linear models for these continuous %C and %N data is more appropriate than creating boxplots of artificially selected bins for comparison. If the reviewer does not agree with this way of handling these data, we are happy to provide the suggested box plot but identifying data bins is somewhat subjective, and would result in different numbers of observations in different bins. We hope the reviewer agrees with our decision to base the analysis on the full dataset.

r164: if $p=0.24$, how is this statistically significant?

This was a typo. After re-running the statistical tests and double checking the outputs it turns out that the RPE ratio was significantly positively related to C:N ratio in active-layer soils ($R^2=0.38$, $p=0.024$). We have corrected this error in the manuscript text.

r188-189: 'may alleviate RPEs': What do you mean with this? One could understand 'may alleviate microbial N-limitation, leading to lower N-mining in SOM and resulting in the absence of a priming effect'. However, more likely than a microbial N limitation, especially in soils that had a thaw-related nutrient flush, is a plant N limitation and a microbial C limitation, which is explained nicely in this paper (Soong et al., 2020). This assumption would lead to an entirely different interpretation of the results. Moreover, no evidence or measurements are given for the statement that the available nutrient pool diminished after 185 days. The initial absence of RPE and recovery later on might be induced by other dynamics as well, like a N-induced shift in microbial community rather than RPEs being driven by nutrient demand.

The reviewer makes some excellent points here. We have expanded this section of the manuscript to include some of these suggestions as well as improve upon the discussion of the nutrient addition data and how this links to the data from soils without nutrient addition (L316-358).

r215-228: It is true that little is known about SOM formation and stabilisation in high-latitude areas, but this is not what your experiment or research is about. Maybe the MS would benefit more from a deeper interpretation of your (interesting!) results than from this digression.

We have removed this section from the manuscript as per the reviewer's suggestion.

r459-469: Did you test your data for within-group normality (t-test) or normality of the residuals (ANOVA). It's not mentioned here, but remains an important assumption.

Within group and residual normality was confirmed using the Shapiro-Wilk test. Text stating this has been added to the methods section (L621-622).

r475: Please have a look at the notation of this equation. Mention units consistently (60 min h⁻¹), mend spaces (mlmol⁻¹), write physical quantities and units the same for each term (g C), and take in account conversions (g → μg)

We have double-checked the notation of the equation in question and made relevant edits as necessary (L636-639).

r476: Why FLOWair is capitalised when it's not in the equation?

This error has been corrected in the text, so that the equation and text now match (L640).

r480: I don't understand why the RPE ratio is noted like $\exp(\ln(\text{CO}_2\text{fluxroots}/\text{CO}_2\text{fluxnoroots}))$, which just comes down to $\text{CO}_2\text{fluxroots}/\text{CO}_2\text{fluxnoroots}$ for values other than 0.

This was indeed not very clear in the original text. We have now clarified in the manuscript (L646) that the RPE ratio for each measured time point was calculated by: $\exp(\text{mean}(\ln(\text{CO}_2\text{fluxroots}/\text{CO}_2\text{fluxnoroots})))$. It is essential to use natural logs when averaging ratios, otherwise average ratios greater than one are likely to be observed even when there is no overall response. This is analogous to the approach used in meta analyses.

Supplementary

r24: moth → month

Corrected in text.

Müller, R., Maier, A., Inselsbacher, E., Peticzka, R., Wang, G., & Glatzel, S. (2022). ¹³C-Labeled Artificial Root Exudates Are Immediately Respired in a Peat Mesocosm Study. *Diversity*, 14(9), Article 9. <https://doi.org/10.3390/d14090735>

Soong, J. L., Fuchslueger, L., Marañon-Jimenez, S., Torn, M. S., Janssens, I. A., Penuelas, J., & Richter, A. (2020). Microbial carbon limitation: The need for integrating microorganisms into our understanding of ecosystem carbon cycling. *Global Change Biology*, 26(4), 1953–1961. <https://doi.org/10.1111/gcb.14962>

We thank both reviewers for their helpful comments and final revisions.

Responses to the reviewers comments are in blue below:

REVIEWERS' COMMENTS

Reviewer #1 (Remarks to the Author):

I appreciate the authors for their great efforts to address my previous comments. Compared with earlier studies available in permafrost regions, this study added valuable insights into priming effects by conducting a long-term labeling mesocosm experiment (with real plants, climate-controlled growth and continuous C13 labeling chamber experiment). Several remaining minor points need to be addressed.

(1) Line 101: "Results" section should be replaced with "Results and discussion", since the authors also discussed the related results in this section.

We have changed this in the manuscript text.

(2) Line 110: To improve the readability, it would be better to briefly mention the labeling mesocosm experiment before the detailed description of the results, given the detailed methods occurred at the end of the paper.

We thank the reviewer for this suggestion and have added the following text (L109-113):

"The collected permafrost and active layer soils were used to investigate rhizosphere priming effects in permafrost region soils within a temperature controlled, ¹³C-labelling plant-growth chamber²⁷. By measuring soil C effluxes from the headspace of both rooted and root-free compartments in custom-designed mesocosms (see Methods for details), we were able to partition C effluxes into plant-derived C and RPE-induced release of soil C."

(3) Figure 1 and the corresponding results section: Based on this figure, we can see that the priming effect was significant in only two incubation days for the active layer (Fig.1c). Similarly, the priming effect was not significant for all of incubation days in the permafrost (Fig.1c). The authors needed to clarify this point and discuss more for this issue.

Text to emphasise this has been added in L137-138 and L151-152.

(4) Line 170: This argument could not be obtained directly from Fig.2d. It would be better to add an inserted panel for showing this comparison between permafrost and active layer.

We have added an insert to Figure 2d as per the reviewer's suggestion.

(5) Line 235: the reduction of microbial nutrient limitation.

This has been corrected in the manuscript text.

(6) Line 247-248: Could the authors conduct some kind of statistical comparison to illustrate this point in a more direct manner?

We have performed a paired t-test and amended the relevant text to include the results.

Reviewer #2 (Remarks to the Author):

Nina L. Friggens and colleagues have edited their manuscript thoroughly and made an effort to respond to my comments. Most of my questions or comments have been well addressed, although some experimental issues cannot be dealt with without repeating the experiment, such as the low temporal resolution of the sampling, or the lack of destructive sampling to get more insight into soil nutrient dynamics, microbial biomass or community. Although I still consider this to be a weakness of the manuscript, which inevitably leads to speculation in the discussion of plant or microbial nutrient status and what exactly causes the RPE, I agree with the authors in their reasoning that the lack of this information does not necessarily weaken their main conclusion. They have also added, as I requested, a better contextualisation of their results in the discussion, and they also now have a very good, relevant and important conclusion.

r307: Here you mention N-mining as a plant-driven process, but it's a microbial process. Also, microbial N-mining causing positive priming (r54: 'This implies that RPEs are controlled by microbial nutrient demand') is controversial and has been challenged in the literature, so this statement should be nuanced and positive priming should not be attributed to N-mining without reservation (Verbrigghe et al., 2022; Wild et al., 2019).

r307: We have amended the manuscript text to avoid suggesting that N-mining is a plant-driven process.

r54: We thank the reviewer for the suggested references which do indeed add nuance to the context of the role of N-mining in positive priming effects. We have amended the text in L53-54 to read; "This implies that RPEs may be controlled by microbial nutrient demand, however evidence from incubation experiments suggests that microbial N-mining may not be a key driver of positive priming."

Verbrigghe, N., Meeran, K., Bahn, M., Fuchslueger, L., Janssens, I. A., Richter, A., Sigurdsson, B. D., Soong, J. L., & Vicca, S. (2022). Negative priming of soil organic matter following long-term in situ warming of sub-arctic soils. *Geoderma*, 410, 115652. <https://doi.org/10.1016/j.geoderma.2021.115652>

Wild, B., Li, J., Pihlblad, J., Bengtson, P., & Rütting, T. (2019). Decoupling of priming and microbial N mining during a short-term soil incubation. *Soil Biology and Biochemistry*, 129, 71–79. <https://doi.org/10.1016/j.soilbio.2018.11.014>